# Chinese Lexical Substitution: Dataset and Method

**Jipeng Qiang** [1] and **Kang Liu**[1] and **Ying Li**[1] and **Yun Li**[1] and **Yi Zhu**[1]
and **Yunhao Yuan** [1] and **Xiaocheng Hu** [2] and **Xiaoye Ouyang**[2] [*]

[1]School of Information Engineering, Yangzhou University, Yangzhou, China

[2]China Academy of Electronic and Information Technology, Beijing 100041, China

{jpqiang,liyun, yhyuan, zhuyi}@yzu.edu.cn,
{liukang,liying}@stu.yzu.edu.cn, ouyangxiaoye@cetc.com.cn

## Abstract

Existing lexical substitution (LS) benchmarks were collected by asking human annotators to think of substitutes from memory, resulting in benchmarks with limited coverage and relatively small scales. To overcome this problem, we propose a novel annotation method to construct an LS dataset based on human and machine collaboration. Based on our annotation method, we construct the first Chinese LS dataset CHNLS which consists of 33,695 instances and 144,708 substitutes, covering three text genres (News, Novel, and Wikipedia). Specifically, we first combine four unsupervised LS methods as an ensemble method to generate the candidate substitutes, and then let human annotators judge these candidates or add new ones. This collaborative process combines the diversity of machine-generated substitutes with the expertise of human annotators. Experimental results that the ensemble method outperforms other LS methods. To our best knowledge, this is the first study for the Chinese LS task.

## 1 Introduction

Lexical substitution (LS) aims at finding appropriate substitutes for a target word in a sentence, which can be used as a backbone of various NLP applications such as writing assistance (Lee et al., 2021; Qiang et al., 2023a), word sense disambiguation (McCarthy, 2002), and lexical simplification (Paetzold and Specia, 2016; Qiang et al., 2021a,b). For instance, when presented with the sentence "I read an amazing paper today", we aim to select a more descriptive adjective to substitute the word "amazing". While options such as "awesome" and "great" may readily come to mind, it proves arduous for us to conceive of equally fitting alternatives such as "incredible" and "fascinating". Despite extensive research conducted on Lexical Substitution (LS) in various languages, including

---

[*]Corresponding author.

English (Hassan et al., 2007; Yuret, 2007; Melamud et al., 2015b; Lee et al., 2021; Qiang et al., 2023b), German (Hintz and Biemann, 2015, 2016), Italian (Toral, 2009), and Croatian (Alagić and Šnajder, 2017), Chinese LS has received limited attention. In this paper, we address this gap by focusing on the Chinese LS task.

To enable the development and evaluation of effective Chinese LS methods, a large-scale dataset is intuitively important. Existing widely used English LS benchmarks, LS07 (McCarthy and Navigli, 2007), CoInCo (Kremer et al., 2014), and SwordS (Lee et al., 2021), were collected by asking human annotators to think of substitutes from memory. The annotation method has the following two problems.

(1) Limited Coverage: Human annotators may have limitations in recalling a comprehensive range of potential substitutes for a given target word, potentially overlooking less common or domain-specific substitutes (Liu et al., 2022). Much work (Lee et al., 2021; Qiang et al., 2023b) have also pointed out the lack of coverage of existing LS datasets. For example, the data collection strategy used in the existing benchmarks might contain words like "awesome" and "great", but miss words like "incredible" and "fascinating".

(2) High cost: Annotating lexical substitutes for target words in sentences is a time-consuming and labor-intensive task. It requires human annotators to carefully consider suitable substitutes, taking into account various linguistic and contextual factors. Due to the complexity of the task, annotating a large number of instances becomes challenging within a reasonable timeframe and budget. Consequently, widely used English LS datasets such as LS07, CoInCo, and SwordS comprise a mere 2,010, 2,474, and 1,250 instances, respectively.

To address these challenges, we propose a novel annotation method to construct an LS dataset based on human and machine collaboration. Firstly, we

propose an ensemble method that leverages four different unsupervised LS methods to automatically generate substitutes. Automated methods can quickly generate a vast pool of potential substitutes, reducing the burden on human annotators. Secondly, we let human annotators assess the suitability of these alternatives as substitutes. Additionally, we request annotators to suggest new alternatives that are not present in the machine-generated options. This collaborative process harnesses the expertise of human annotators while leveraging the efficiency and scalability of machine-generated candidates. This efficiency allows for the creation of a larger dataset within a reasonable budget.

The annotation method is motivated by the following two findings:

(1) Machine-generated LS methods can introduce a greater diversity of substitutes. By leveraging computational techniques like word embeddings, language models, or paraphrasing models, a wide range of plausible substitutes can be generated. This diversity enriches the dataset by providing a variety of substitution options, capturing different semantic relationships and syntactic patterns.

(2) Assessing the suitability of these substitutes is much simpler for the annotator compared to generating a substitute from memory. Human annotators can focus on selecting the most appropriate substitutes from the machine-generated pool, ensuring high-quality and contextually relevant annotations.

In summary, our contributions are listed below:

(1) We provide a novel approach to construct an LS dataset based on human and machine collaboration. Our approach provides a good idea for constructing large-scale, high-coverage LS datasets. Based on our designing method, we construct the first large-scale Chinese LS dataset CHNLS that consists of 33,695 instances, which cover different text genres, namely News, Novel, and Wikipedia articles. Correspondingly, the latest English LS dataset only contains 1,250 instances.

(2) We present four Chinese LS methods (dictionary-based, embedding-based, BERT-based, and paraphraser-based) by adjusting current English LS methods, and give an ensemble method that combines the four methods. Experimental results on CHNLS show that the ensemble method can be served as a strong baseline for future studies.

The dataset and code is available at github [1].

## 2   Related Work

**Lexical Substitution Resources.** Existing lexical substitution (LS) datasets are available for various languages, including English and other languages. Each instance in LS dataset is composed of a sentence, a target word, and corresponding substitutes.

In English, the first LS dataset from SemEval 2007 (LS07) (McCarthy and Navigli, 2007), consists of 300 development and 1,710 test instances for 201 polysemous words. For each target word, 10 sentences are provided. The annotators' task deployed by Amazon Mechanical Turk was to give up to 3 possible substitutes. Afterward, Biemnann (Biemann, 2012) created a large-scale dataset (TWSI) that annotates 25K sentences from Wikipedia, which, however, only covers noun targets. To alleviate this limitation, Kremer et al. (Kremer et al., 2014) proposed Concept In Context (ConInCo), a dataset of 2,474 sentences covering 3,874 distinct targets with different part-of-speech tags, which is the current largest LS benchmark. It consists of 15K target instances with a given 35% development and 65% test. Recently, Stanford Word Substitution Benchmark (SwordS) (Lee et al., 2021) is built on CoInCo by asking human annotators for higher coverage and higher quality. SwordS consists of 1250 instances with a given 417 development and 833 test. Considering the size of vocabulary in English, the size of the vocabulary covered by LS datasets is too small. Additionally, we found that many appropriate substitutes for many instances in SwordS are missing, since human annotators frequently utilize repetitive patterns to fabricate instances, leading to a lack of linguistic diversity (Liu et al., 2022).

The German LS dataset from GermEval 2015 consists of 2,040 instances from the German Wikipedia, which contains 153 unique target words. Italian LS dataset from EVALITA 2009 consists of 2,310 instances, which contains 231 unique target words. All the above LS datasets in all languages are constructed by human annotators. Due to their relatively small size, all of these datasets can only be used for evaluation and not for training. Unfortunately, research on Chinese LS is still scarce: to the best of our knowledge, there is currently no publicly available LS corpora for training, even lacking a dataset to evaluate the ability of LS models.

---

[1]https://github.com/KpKqwq/CHLS

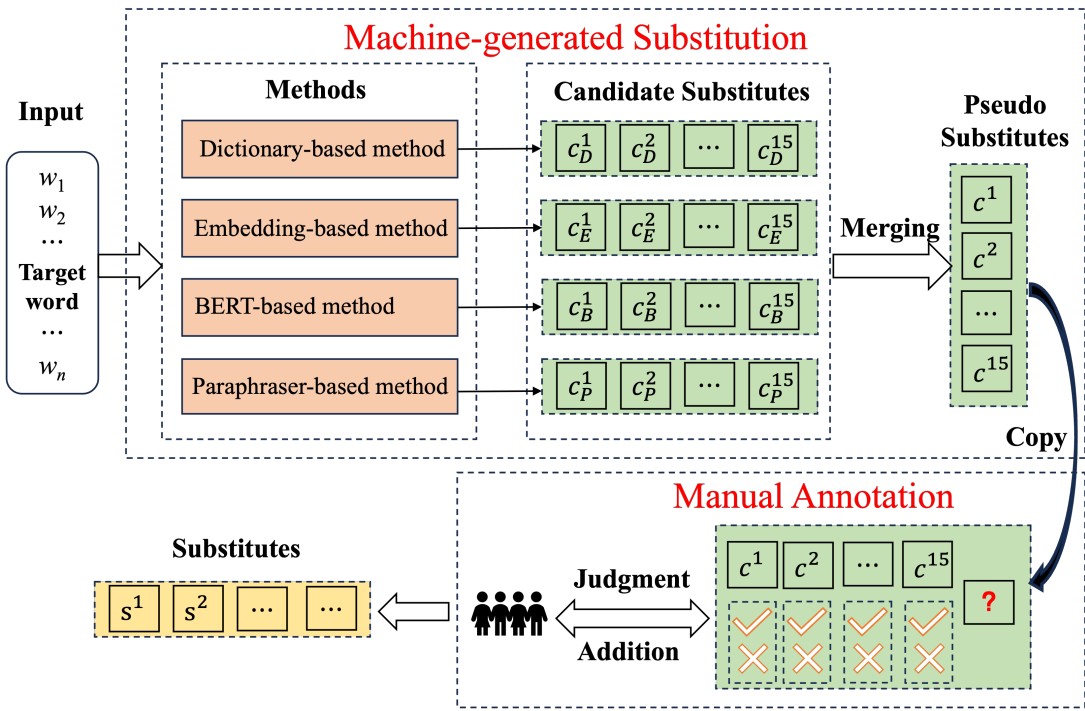

Figure 1: The overview of our approach for building Chinese LS corpus. Our approach is composed of two phrases: machine-generated substitution and manual annotation. The phase of machine-generated substitution combines four different LS methods as an ensemble method to generate the pseudo substitutes. The phase of manual annotation utilizes native Chinese annotators to judge the pseudo substitutes and add new substitutes.

**Lexical Substitution.** LS methods can be divided into four types: (1) dictionary-based method (Hassan et al., 2007; Yuret, 2007), (2) Embedding-based method (Melamud et al., 2015a,b), (3) BERT-based method (Zhou et al., 2019; Lacerra et al., 2021a; Michalopoulos et al., 2022), and (4) Paraphraser-based method (Qiang et al., 2023c,b).

The early lexical substitution studies obtain synonyms by searching linguistic resources, such as WordNet. Embedding-based methods utilize word embedding modelings to obtain highly similar words as the substitutions. Since 2019, LS methods based on pretrained language models have attracted much attention (Zhou et al., 2019; Lacerra et al., 2021a; Michalopoulos et al., 2022), in which pretrained BERT is most used. Zhou et al. (Zhou et al., 2019) apply dropout to the embeddings of target words for partially obscuring the word, and obtain a probability distribution over the BERT output vocabulary. Arefyev et al. (Arefyev et al., 2020) present a comparative study of popular pretrained language models, such as ELMo, BERT, and XL-Net. Lacerra et al. (Lacerra et al., 2021b) first merge the development set of two LS datasets (Co-InCo and TWSI), and split it into training and development sets for fine-tuning the encoder-decoder

framework. Michalopoulos et al. (Michalopoulos et al., 2022) propose a new mix-up embedding strategy by incorporating the knowledge of Word-Net into the prediction process of BERT. Recently, Qiang et al (Qiang et al., 2023b) propose a method ParaLS that utilizes a pretrained paraphraser to generate the substitutes. Compared to language modeling, paraphraser produces fluent, meaning-preserving paraphrases but contain variations in word choice. ParaLS achieves good results and is considered the state-out-of-art LS method.

## 3 Creating CHNLS

In this section, we describe our method to build an LS dataset, and the overall architecture for constructing the Chinese LS corpus is illustrated in Figure 1.

### 3.1 Data Preparation

In this step, we extract the sentences and the corresponding target words. To ensure diversity and complexity in our dataset, we utilize three distinct text genres: News, Novel, and Wiki. The News category is sourced from the contents of People's Daily, Wiki consists of articles from Wikipedia (encyclopedia), and the Novel category comprises

selected Chinese-published novels. By incorporating multiple sources, we aim to capture the richness and intricacy of the Chinese language.

To refine the dataset, we apply a filtering process to eliminate excessively short or long sentences based on their word count. For each sentence, we further segment it into words, considering nouns, verbs, adjectives, and adverbs as the target words for our analysis.

## 3.2 Machine-generated Substitution

Considering the sentence $w_1, w_2, ..., tw, ..., w_n$ containing the target word $tw$, we employ LS methods to generate a set of 15 pseudo substitutes for each target word. To foster a broader range of substitutes, we adopt an ensemble approach that combines four distinct LS methods: Dictionary-based, Embedding-based, BERT-based, and Paraphraser-based. By leveraging these diverse methods, each of which taps into different semantic knowledge, we aim to enhance the overall diversity of substitutes available for consideration.

Typically, LS methods encompass two essential steps: substitute generation and substitute ranking. The initial objective of substitute generation is to identify and produce potential alternatives that can effectively replace a target word within a given sentence. Once a set of substitute candidates is generated, the subsequent task of substitute ranking comes into play, aiming to ascertain the most appropriate substitute for the target word within the specific sentence.

**Substitute Generation.** We present four baseline approaches by adapting existing English LS methods:

(1) Dict-based: The dictionary-based method relies on a synonym thesaurus (HIT-Cilin (Mei et al., 1996)) to generate the candidate substitutes.

(2) Embedding-based: The embedding-based method selects substitutes with the highest similarities from word embedding models (Li et al., 2018). Substitutes are chosen based on their proximity, as determined by cosine similarity, to the target word.

(3) BERT-based: The BERT-based method (Qiang et al., 2021b) utilizes Chinese BERT modeling[2] and masks the target word for prediction.

(4) Paraphraser-based: The Paraphraser-based method (Qiang et al., 2023b) leverages a pretrained paraphrase model to generate substitutes. By inputting the sentence into the encoder of the para-

phrase model, substitutes are generated using a special decoding strategy that focuses exclusively on the lexical variations of the target word.

Given the absence of a suitable Chinese paraphraser and a sufficiently large-scale paraphrase corpus, we take the initiative to construct a comprehensive Chinese paraphrase corpus. This corpus is then utilized to fine-tune a pretrained Chinese BART model[3], enhancing its effectiveness for paraphrase generation.

To construct a paraphrase corpus, we leverage a large-scale bilingual English-Chinese translation corpus. The construction process entails the following steps:

(1) Gathering the machine translation corpus: We select a Chinese-English corpus consisting of 5.2 million sentence pairs[4] as our primary source.

(2) Aligning sentence pairs: We utilize a Chinese translator[5] to translate the English sentences into Chinese, thus creating aligned sentence pairs representing paraphrases.

(3) Identifying potential paraphrases: By comparing the aligned sentence pairs, we identify pairs that convey similar or identical meanings while being expressed differently. These pairs serve as potential paraphrases.

(4) Filtering and cleaning paraphrase pairs: We apply filters to remove unsuitable sentence pairs for paraphrase generation. For instance, we exclude pairs with significant length differences, pairs containing mistranslations, or pairs exhibiting inconsistencies.

Through these steps, we construct a high-quality paraphrase corpus that can be used for various natural language processing tasks, including paraphrase generation and LS.

**Substitute Ranking.** The effectiveness of text generation metrics for substitute ranking has been demonstrated in previous work (Qiang et al., 2023b). Therefore, we employ the BARTScore (Yuan et al., 2021) and BERTScore (Zhang et al., 2019) metrics for substitute ranking. To perform this ranking, we replace the target word in the original sentence with each substitute candidate, thereby creating an updated version of the sentence.

BARTScore leverages pre-trained BART models to calculate the similarity between the original sentence and the updated sentence. BARTScore

---

[2]https://huggingface.co/bert-base-chinese

[3]https://huggingface.co/fnlp/bart-base-chinese
[4]https://github.com/brightmart/nlp$_c hinese_c orpus$
[5]https://huggingface.co/Helsinki-NLP/opus-mt-en-zh

considers various aspects of text quality, including fluency, grammaticality, and semantic similarity.

BERTScore utilizes pre-trained BERT models to measure the similarity between the original sentence and the updated sentence. BERTScore has shown a strong correlation with human judgments and has been widely used for evaluating text generation tasks.

Finally, our ranking method employs a linear combination of the scores of BARTScore and BERTScore to compute the final score for each candidate substitute. They consider different aspects of text quality and provide comprehensive measures to rank the substitutes based on their similarity to the reference word. By incorporating these metrics, the ranking process can be more robust and accurate, leading to an improved selection of suitable substitutes in lexical substitution tasks.

**A ensemble Method.** The aforementioned four LS methods utilize substitute generate and substitute ranking to generate 15 substitutes separately for each method. Specifically, the substitutes generated by Dictionary-based, Embedding-based, BERT-based, and paraphraser-based methods are denoted as $\{c_D^1, ..., c_D^{15}\}$, $\{c_E^1, ..., c_E^{15}\}$, $\{c_B^1, ..., c_B^{15}\}$, and $\{c_P^1, ..., c_P^{15}\}$, as shown in Figure 1.

Taking into consideration that each LS method generates 15 substitutes, the utilization of four LS methods results in a total of 60 candidate substitutes. To avoid overwhelming the annotators and incurring additional costs, as well as to prevent annotator fatigue, we need to limit the number of substitutes for annotation.

To achieve this, we propose a simple ensemble method that combines the above four methods. We assigned voting weights of 1 to Dict-based, Embedding-based, BERT-based, and paraphraser-based methods individually. We select the top 15 candidate substitutes with the highest votes, denoted as $\{c^1, c^2, ..., c^{15}\}$, as pseudo substitutes. This selection process ensures that the substitutes generated by multiple methods are more likely to be chosen as potential substitutes.

### 3.3 Manual Annotation

Given the sentence and target word pairs, as well as the corresponding 15 pseudo substitutes $\{c^1, c^2, ..., c^{15}\}$, we engage multiple annotators for annotation. It is noteworthy that all the annotators involved in this process are native Chinese under-

|  | **Wiki** | **Novel** | **News** | **Total** |
|---|---|---|---|---|
| **Insts.** | 10,378 | 11,566 | 11,751 | 33,695 |
| **Sents** | 1,111 | 1,138 | 1,284 | 3,533 |
| **Subs.** | 48,251 | 48,225 | 48,232 | 144,708 |
| **Avg.** | 4.1 | 4.8 | 5.9 | 4.9 |

Table 1: The statistics of the constructed Chinese LS dataset. "Insts." is the number of instances, "Sents." is the number of different sentences, "Subs." is the total number of substitutes, and "Avg." is the average number of substitutes per target.

graduates.

We have created a specialized website for annotating data. On each page of the website, a sentence is presented with a highlighted target word, along with 15 pseudo substitutes for that target word. Additionally, there is an option to add new substitutes that are not included among the pseudo-substitutes. For each pseudo substitute, there are two radio buttons labeled "positive" and "negative." The annotators' task was to select "positive" if they considered the substitute to be a suitable replacement for the target word within the given sentence. Conversely, they were to choose "negative" if they determined that the substitute would not be appropriate.

To encourage annotators to contribute new substitutes, we offer higher compensation for providing new substitutes that are not included among the pseudo-substitutes. During the annotation process, each sentence and target word pair in the dataset is annotated three times. The final substitutes are selected from the newly added substitutes and the pseudo-substitutes that have been marked at least once.

We conducted a pilot test with one annotator, and they were able to annotate approximately 150 instances in one hour. The average time required per assignment was approximately 25 seconds, which may seem surprising. However, two factors contribute to this efficiency: (1) Native speakers can quickly make binary judgments regarding substitute words. (2) Annotators only need to read the target sentence once to provide judgments for all substitutes in an assignment. For more information on the interface, instructions, and filtering criteria, please refer to Appendix A.

## 4 Data analysis

The statistical information of the constructed Chinese LS dataset, CHNLS, is presented in Table 1.

The dataset consists of a total of 33,695 sentences and target word pairs, with a corresponding 144,708 labeled substitutes. On average, close to 10 words per sentence are selected as target words. We calucate named as

**High quality.** The objective is to evaluate the accuracy of the substitutions made in the given sentence and target word pairs. A total of 300 instances were randomly selected, with 100 instances chosen from one of three text genres. A new annotator, proficient in the Chinese language, was assigned the task of assessing the precision of the substitutions within the selected instances.

This annotator compared each substitute against the original target word to determine if it accurately captured the intended meaning and maintained syntactic and semantic coherence within the sentence. He classified the substitutions as correct or incorrect. The precision of the substitutions was computed by dividing the number of correct substitutes by the total number of substitutes evaluated. The precision would be calculated as 1136/1254, which is equivalent to 90.5%. The high precision rate of above 90% indicates the high quality of the substitutions within the dataset.

**High coverage.** We show that CHNLS achieves high coverage. The same 300 instances in high quality are selected. Three new human evaluators, proficient in the Chinese language, were asked to independently think of substitutes for each sentence and target word pair in the selected instances.

The substitutes provided by the evaluators are compared against the set of substitutions present in the constructed dataset. Each substitute is evaluated to determine whether it matched any of the substitutions in the dataset. The coverage of the dataset is calculated by dividing the number of substitutions provided by the human annotators that belonged to the dataset's set of substitutions by the total number of substitutions provided.

The human annotators provide 742 substitutions and 723 substitutions belonged to the substitutions provided in the CHNLS. The coverage is calculated as 723/742, which is equivalent to 97%. This verification process demonstrates the extensive coverage of the dataset and its suitability for training and evaluating Chinese LS models. Additionally, it is worth noting that the three annotations only yielded a total of 742 substitutes, which is significantly smaller than the 1254 substitutes present in the dataset. This observation highlights the imprac-

| Cohen's (A1–A2) | Cohen's (A1–A3) | Cohen's (A2–A3) | Fleiss' kappa |
|---|---|---|---|
| hline 0.598 | 0.614 | 0.572 | 0.594 |

Table 2: Cohen's kappa agreement scores for pairs of annotators and Fleiss' kappa agreement for three annotators

ticality of relying solely on manual annotation for generating language substitution word data, as it results in a substantial lack of coverage.

**High agreement.** We used common agreement metrics such as Cohen's Kappa(Cohen, 1960) and Fleiss' Kappa(Fleiss, 1971) to quantify the level of agreement among annotators. Cohen's Kappa measures agreement between two raters, and Fleiss' Kappa can be used for measuring agreement between multiple raters. The Kappa result be interpreted as follows: values $\leq 0$ as indicating no agreement and 0.01–0.20 as none to slight, 0.21–0.40 as fair, 0.41–0.60 as moderate, 0.61–0.80 as substantial, and 0.81–1.00 as almost perfect agreement.

Table 2 lists the agreement scores for three annotators. Specifically, we calculated Fleiss' Kappa for our dataset, yielding a value of 0.594. This statistic underscores a substantial level of agreement among our human annotators, reaffirming the consistency and reliability of the annotations.

## 5 Experiments

### 5.1 Experimental Setup

**Dataset.** We split the whole dataset CHNLS into train (80%), valid (10%), test (10%) set. The train/valid/test sets in Wiki, News, and Novel have 8,425/1,065/888, 9,472/1,169/1,110, and 9,379/1,080/11,07 instances, respectively. The experimental results are computed on test sets.

**Metrics.** We employ the designated official metrics, namely "best," "best-m," "oot," and "oot-m," as outlined in the SemEval 2007 task. In addition, we incorporate Precision@1 (P@1) as an evaluation metric, adhering to the conventions established by previous LS methodologies (Zhang et al., 2019; Qiang et al., 2023b). Notably, "best," "best-m," and "P@1" gauge the quality of the most accurate predictions, while both "oot" (out-of-ten) and "oot-m" assess the extent to which the gold substitutes is encompassed within the top 10 predictions.

**Implementation Details.** Dict-based(Dict), Embedding-based (Embedding), and BERT-based (BERT): we use the default settings of the pretrained modeling during constructing our LS datasets. For Paraphraser-based(ParaLS), we first

| Data set | Method | best | best-m | oot | oot-m | P@1 |
|---|---|---|---|---|---|---|
| Wiki | ChatGLM | 12.2 | 17.9 | 34.1 | 39.3 | 35.4 |
| | ChatGPT | 15.8 | 26.1 | 39.4 | 47.5 | 44.5 |
| | Dict | 16.4 | 24.0 | 39.0 | 41.4 | 51.2 |
| | Embedding | 13.6(1.6) | 21.2(1.4) | 46.4(18.5) | 55.1(22.1) | 41.1(0.1) |
| | BERT | 16.9(1.8) | 25.7(2.6) | 53.9(37.1) | 58.4(47.5) | 50.0 (0.1) |
| | ParaLS | 17.0(11.2) | 25.7(16.2) | 54.1(44.1) | 59.8(49.9) | 48.5(36.5) |
| | Ensemble | **18.7** | **29.4** | **66.6** | **73.4** | **57.0** |
| News | ChatGLM | 9.9 | 15.5 | 33.8 | 42.1 | 48.1 |
| | ChatGPT | 14.0 | 28.3 | 36.6 | 43.5 | 62.0 |
| | Dict | 13.2 | 24.5 | 37.3 | 41.6 | 65.9 |
| | Embedding | 12.9(2.0) | 23.6(3.8) | 45.5(16.1) | 56.5(16.9) | 55.8(0.1) |
| | BERT | 17.2 (1.2) | 32.9(1.4) | 62.1(38.8) | 66.6(49.5) | 80.9(0.1) |
| | ParaLS | 17.3(11.4) | 34.5(22.6) | 66.3(51.5) | 71.2(58.4) | 79.5(64.6) |
| | Ensemble | **18.36** | **36.68** | **77.20** | **87.77** | **84.86** |
| Novel | ChatGLM | 10.3 | 11.0 | 32.3 | 34.0 | 38.1 |
| | ChatGPT | 12.4 | 14.7 | 35.4 | 37.5 | 45.5 |
| | Dict | 12.5 | 17.0 | 41.3 | 40.0 | 54.5 |
| | Embedding | 14.0(1.7) | 16.8(1.8) | 46.5(18.7) | 46.4(20.0) | 46.1(0.1) |
| | BERT | 13.8(0.9) | 18.6(1.4) | 44.9(38.5) | 47.8(45.5 ) | 54.2(0.1) |
| | ParaLS | 13.8(7.3) | 18.6(12.2) | 52.8(39.0) | 60.0(40.7) | 53.4(42.4) |
| | Ensemble | **18.12** | **26.67** | **65.83** | **70.34** | **64.31** |

Table 3: Evaluation results of substitute generation and substitute ranking. The scores in parentheses are only calculated by the substitutes from the substitute generation step. The Best values are bolded and the second values are marked in blue.

construct a large Chinese paraphrase dataset, containing 5,174,152 sentence pairs. Then we fine-tune Chinese BART on it to train a paraphraser. The initial learning rate is set to $lr = 1 \times 10^{-5}$ and dropout is set to 0.1. We adopt the Adam optimizer with $\beta_1 = 0.9$, $\beta_2 = 0.999$, $\epsilon = 10^{-8}$. For the above methods, we set the max number of the generated candidates as 50. We use BARTscore and BERTscore to rank the candidates and select the top 10 words to calculate metrics. The weights are set as 1, 0.1 for BARTscore and BERTscore for all the above methods. To validate vLLM's ability on this dataset, we also tested two LLMs: ChatGLM[6] and ChatGPT[7],using their official API interfaces.

## 5.2 Evaluation Results

Table 3 displays the performance of all methods on the various metrics. To eliminate the impact of substitute ranking, we also provide the results without substitute ranking in parentheses.

Among the individual methods, we observed that BERT and ParaLS outperform the baselines

[6]https://open.bigmodel.cn/
[7]https://platform.openai.com/

Dict and Embedding. This is because both BERT and ParaLS utilize pretrained models that incorporate contextual information for better predictions. Without substitute ranking, ParaLS achieves better performance than BERT. It also means that ParaLS based on our constructed paraphrase corpus is the best individual LS method. When compared with vLLMs, we found BERT and ParaLS also outperform ChatGPT and ChatGLM.

Experimental results demonstrate that our proposed method Ensemble surpasses the individual LS methods on all metrics with statistical significance. Ensemble expands the coverage of possible substitutes by utilizing multiple LS methods. Each method has its own coverage limitations and biases. By combining them, Ensemble overcomes individual limitations and biases, leading to broader coverage of substitute candidates. This broader coverage increases the likelihood of finding suitable substitutes for a wide range of target words. Additionally, different LS methods may exhibit varying levels of sensitivity to different linguistic contexts, word senses, or syntactic structures. By combining multiple methods, the ensemble approach becomes

| | |
|---|---|
| **Inst. 1** | 普乌土斯克陨石是包含镶入两种变体的碎性砾石的暗黑色砾石基石。 |
| English | The Pułtusk meteorite is a dark black conglomerate containing crushed gravels set into two variants. |
| Labels | 含有;涵盖;包括;蕴含;蕴涵;具有;富含;囊括;富有 |
| Dict | 含有;含(contain);带有(have);蕴含;富含;涵盖;隐含(implication);饱含(full of);包孕(include);蕴藏(hold in store) |
| Embedding | 括(include);蕴含着(contains);蕴涵;具备(with);收录(inclusion);隐含(implict);所含;包括;涵括(include);程序包(program Packages) |
| Bert | 含有;含(contains);包括;一块(one piece);具有;一种;一个;拥有;富含;内含 |
| ParaLS | 含有;含;包括;一块;带有;具有;一种(a kind of);一颗(one);带(with);蕴含 |
| Ensemble | 含有;含(contains);包括;带有(with);蕴含;富含;涵盖;一块(one piece);具有;一种(a kind of) |
| **Inst. 2** | 张魁，本名张毅民，是谐星张帝的弟弟，以歌曲〈小丑〉成名。 |
| English | Zhang Kui, whose real name is Zhang Yimin, is the younger brother of the harmonizer Zhang Di, who became famous with the song "Clown". |
| Labels | 出名;扬名;走红;蜚声;名气;有名;名满天下 |
| Dict | 出名;一举成名(one shot at fame);名声大振(gaining fame);著称(be known as);名扬四海(famous all over the world);扬名;名声鹊起(Rise to fame); textcolorred走红;一鸣惊人(make a great coup);驰名(well-known) |
| Embedding | 名声大噪(gain fame and notoriety);风靡(popular);一片成名(A piece of fame);蹿红(leap into the limelight);一炮而红(a hit); 成名作(famous works);功成名就者(those who have achieved fame and fortune);名声大振(gaining fame);大红大紫(hit the big time); 功成名就(success and fame) |
| Bert | 出名;闻名(famous);知名(famous);有名;得名;走红;出道(debut);取名(naming);成功(success);起名(name) |
| ParaLS | 出名;闻名(famous);著称(be known as);著名(famous);知名(famous);为名(in the name of);有名;得名(got the name);名声(fame);走红 |
| Ensemble | 出名;走红;扬名;出道(Debut);闻名(famous);一举成名(one shot at fame);著称(be known as);名声大振(gaining fame);知名(Famous);有名 |
| **Inst. 3** | 柠檬酸镁是镁的柠檬酸盐，它通常以水合物的形式存在。 |
| English | Magnesium citrate is the citrate salt of magnesium, which is usually present in the form of a hydrate. |
| Labels | 姿态;样式;格式;方式;类型 |
| Dict | 样式;式子(formulas);式样(style);款式(styles);花样(pattern);式(style);试样(specimen);款型(styles);样款(sample) |
| Embedding | 多形式(multiform);型式(type);传播方式(dissemination method);多种形式(multiple forms);表达方式(expression);不拘形式(unconventional); 新形式(new form);载体(carrier);名目(name);模式(mode) |
| Bert | 方式;形态(morphology);形状(shape);状态(status);样式;身份(identity);型式(type);姿态;类型;模式(mode) |
| ParaLS | 方式;形态(morphology);形状(shape);状态(status);样式;身份(identity);型式(type);姿态;类型;模式(mode) |
| Ensemble | 方式;样式;型式(type);模式(mode);形态(morphology);形状(shape);状态(status);身份(identity);式子(formulas);姿态 |
| **Inst. 4** | 柠檬酸镁可用作生理盐水轻泻剂以及镁的营养来源。 |
| English | Magnesium citrate is used as a light laxative in saline and as a nutritional source of magnesium |
| Labels | 作;当;作为;用做;用来;当作;当做;使用;做;应用 |
| Dict | 作为;作;当作;当做;当;视作(treat as);看作(look upon as);看做(look upon as);看成(regard as);同日而语(talk in the same breath) |
| Embedding | 当做;媒染剂(Mordant);选作(Selected Works);列作(Listed as);主要用途(main Uses);借作(borrow as);用途(Use);用于(For);移作(shift to);用来 |
| Bert | 作为;用做;作;以是(so);成为(become);做为(act as);当作;为(for);用为(used as);制成(made to order) |
| ParaLs | 作为;用于(for);用做;以是(so);作;成为(become);以为(thought);做为(act as);当作;为(for) |
| Ensemble | 作为;当作;用做;作;做为(act as);当做;用为(used as);当;用于(for);以是(so) |

Table 4: The top 10 substitutes of four instances in the Wiki subset of CHNLS using LS methods. The target word is marked in blue, and the substitutes in labels are marked in red.

more robust to such variations, as it can draw on the strengths of different methods to handle different linguistic scenarios effectively. This robustness contributes to the overall improved performance.

These reasons indicate that Ensemble benefits from the diversity, enhanced coverage, and robustness of individual LS methods. The combination of these factors contributes to the significant outperformance of the ensemble approach over the individual LS methods on all evaluation metrics, demonstrating its effectiveness in generating high-quality substitutes for the Chinese LS task.

## 5.3 Qualitative evaluation

To qualitatively evaluate the effectiveness of the substitutes generated by LS methods, we present four instances of the Wiki subset of CHNLS for analysis. Table 4 displays the top 10 generated substitutes. More instances are shown in Appendix B.

It is evident that the substitutes we have annotated exhibit a considerable level of comprehensiveness, without any significant absence of suitable substitutes. This observation indicates the high cov-

erage achieved by our constructed dataset. In comparison, even the latest English lexical substitution datasets, such as SwordS which is the improved version of CoInCo, still exhibit deficiencies in capturing a sufficient number of appropriate substitutes (Qiang et al., 2023b).

Consistent with the findings from the quantitative evaluation, the performance of the Dict-based and Embedding-based methods, which do not take contextual information into account during the substitution generation process, is relatively low compared to other methods.

BERT and ParaLS approaches demonstrate promising results in terms of capturing contextual information and generating semantically similar substitutes. By leveraging the strengths of different approaches, Ensemble has two advantages. Firstly, Ensemble yields a greater number of appropriate alternatives when compared to BERT and ParaLS. Across the five instances, BERT, ParaLS, and Ensemble produce 20, 19, and 24 correct substitutes, respectively. Secondly, certain well-suited alternatives that were initially ranked lower in the individual methods ascend to higher positions. For in-

stance, the substitute "走红" (meaning "to become famous") in instance 2 exhibits a notable elevation, securing the second rank.

## 6 Conclusions

This study presents the first comprehensive exploration of the Chinese Lexical Substitution (LS) task. We propose a novel annotation method to construct a large-scale Chinese LS dataset through a collaborative human-machine approach. The constructed dataset consists of 33,695 instances and 165,105 substitutes with high quality and high coverage. Our proposed ensemble method by leveraging the strengths of each method while mitigating their weaknesses, our ensemble approach significantly outperforms the individual LS methods across all evaluation metrics.

In conclusion, our study fills the research gap on how to construct a large-scale LS dataset with high coverage and low cost, providing a solid foundation for further research and development. The construction of a high-quality dataset and the development of an effective ensemble method showcase the potential for improved lexical substitution in the Chinese language.

## Limitations

While our proposed collaborative approach successfully constructs a large-scale Chinese Lexical Substitution (LS) dataset, it is important to acknowledge some limitations to provide a balanced perspective.

Despite the large-scale nature of the dataset, it may not cover all possible lexical substitution scenarios in the Chinese language. The dataset's coverage might be limited to three genres (Wiki, News, Novel), which could affect its applicability in certain contexts. Researchers should be cautious when generalizing findings beyond the dataset's scope.

While efforts were made to ensure annotator agreement through guidelines and quality control measures, some level of inconsistency in judgments among human annotators is inevitable. The inter-annotator agreement might vary for different instances, which could introduce some noise or ambiguity in the dataset.

## Ethics Statement

The dataset used in our research is constructed using publicly available data sources, ensuring that there are no privacy concerns or violations. We do not collect any personally identifiable information, and all data used in our research is obtained following legal and ethical standards.

An additional ethical concern revolves around the possibility of the Chinese LS method being exploited for malicious intents, including the generation of fabricated or deceptive content. It is imperative to contemplate the potential repercussions arising from the outputs of the LS method and to implement protective measures to deter its exploitation for nefarious objectives.

## Acknowledgement

This research is partially supported by the National Natural Science Foundation of China under grants 62076217, U22B2037 and 61906060, and the Blue Project of Yangzhou University.

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

## A The construction process of CHNLS

We selected three types of corpus (novel, news, and Wikipedia) as the source of the original text. Each document is divided into sentences, and each verb, noun, adjective, and adverb in each sentence

is selected as a potential target word. A total of 12,000 target words were selected from each of the three corpora. Subsequently, we employed four distinct lexical substitution methods to generate a set of 15 candidate words for each target word.

Consequently, every sentence, target word, and corresponding collection of 15 candidate words formed a single sample. Ultimately, we accumulated a comprehensive dataset comprising 36,000 samples. To ensure reliable annotations, each sample was presented to three annotators who were instructed to select appropriate alternatives from the provided word list for tagging. The final annotation results constituted the lexical substitution dataset.

### A.1 Selection of target words

We first divided each type of raw corpus into natural sentences. A natural sentence is a complete sentence that ends with a period, exclamation mark, question mark, or ellipsis and can express a complete meaning. Using a word segmenter, we segment and part-of-speech tag the natural sentences. For each verb, noun, adjective, and adverb in each natural sentence, we select them as potential target words. After removing proper nouns, fixed collocations, and other words that cannot be appropriately substituted, the remaining words are considered target words.

### A.2 Annotation Website

We have built a website based on JavaWeb+MySQL for annotators' labeling work. We provide a portion of the target words and a list of 15 substitute words to three annotators to collect suitable sets of substitute words from them.

To improve the quality of annotation, we have implemented the following three design strategies.

(1) To reduce costs and ensure annotation quality, we adopted a rotating approach to presenting the substitute word lists to annotators. In the annotation of each target word, not all 15 substitute words in the list were provided to a single annotator. Instead, a selective subset of 11 or 12 substitute words was presented. This approach aimed to maintain the quality of annotations by avoiding overwhelming annotators with too many words to annotate at once, while significantly reducing the time required for annotation.

For these 15 words, they were systematically rotated among the four annotators, ensuring equal opportunities for each word to be assigned to an

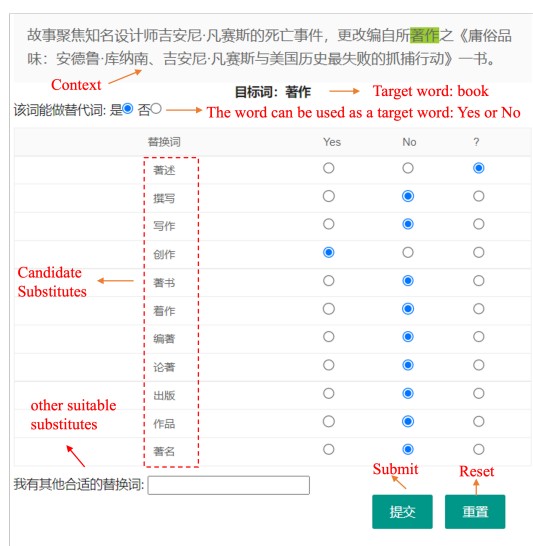

Figure 2: Screenshot of an annotation example on the annotation Website. The red text indicates added comments.

annotator. This rotation strategy does not compromise the reliability of the annotation results, as each word has an equal chance of being assigned to any annotator. Thus, this approach ensures fairness and avoids potential bias in the annotation process.

(2) We modified the substitute word lists for a selected subset of target words and provided them as confusion sets to the annotators to ensure annotation quality. From the original set of 36,000 target words in three corpora, we randomly selected one-third of the target words. For each selected target word, we made modifications to two substitute words out of the 15-word list. One substitute word was changed to the original target word, which served as a required option for the annotators. The other substitute word was replaced with any Chinese word of the same length as the original target word, sourced from a dictionary, and served as a forbidden option for the annotators.

During the annotation process, we evaluated the quality of annotations by checking whether the annotators correctly labeled the confusion set options. This allowed us to assess the annotation quality based on the annotators' handling of the confusion sets.

(3) We have designed three annotation starting positions to ensure consistency in the annotation progress for the three corpora. Each target word has been assigned a unique identifier. Each annotator begins annotating from a designated starting position, which corresponds to the identifier of the target word. To maintain consistency in the an-

notation progress across all corpora, we have established a starting position for annotation at the beginning of each corpus, evenly distributed among multiple annotators.

Once an annotator successfully annotates a target word, the current annotation identifier increments and the next annotatable content is automatically displayed. Only when an annotator reaches the maximum target word identifier, the annotation cycle restarts from the beginning. This approach offers the advantage of enabling consecutive annotations for target words within the same sentence in most cases, effectively reducing the workload of reading sentences, which is the most time-consuming task.

Finally, we eliminated instances that did not contain any meaningful substitute. The number of instances is 33,695.

### A.3 Annotation Manual

This manual is designed to facilitate the use of the Chinese lexical substitution dataset annotation system. It provides instructions on how to use the system effectively and serves as a reference for users, clarifying the purpose and functionality of the system. The manual includes an overview of the task, an explanation of the system's features, specific annotation examples, and a section addressing potential issues that may arise.

### A.4 The Work of Annotators

Annotators are initially instructed to carefully peruse the annotation manual in its entirety. The administrator provides each annotator with a username and password. The administrator also instructs the annotators to annotate the data carefully and explains the website's special features. The system's backend assigns corresponding data to annotators for annotation.

On the annotation website, for each instance, annotators need to determine whether suitable substitute words can be found for the target word in the instance. If an annotator believes that the target word in an instance is not suitable for replacement with any word other than the original word, they can select "Not Replaceable" for that sample and mark all substitute words as "Not Suitable." If an annotator believes that suitable substitute words can be found for the target word in an instance, they need to evaluate and select the appropriate substitute words from the given list. Additionally, annotators can provide alternative suitable substitute words for each instance, different from the ones provided in the given substitute word list. The final collection consists of pairs of target words and the selected substitute word sets as annotated by the annotators.

Regarding the wage for each annotator, our principle is 15¥ per hour. We conducted a pilot test with one annotator, and they were able to annotate approximately 150 instances in one hour. Based on this calculation, the average price for annotating one instance is 0.1¥. To incentivize annotators to provide new words, an additional price of 0.1¥ is offered for each new substitute word.

## B More Examples

Here, we randomly choose 5 instances from News and 5 instances from Novel for analysis in Table 5 and 6.

BERT, ParaLS, and Ensemble provide high coverage and high-quality substitutes compared to Dict and Embedding. These results indicate that Ensemble achieves a little better results.

| Inst. 1 | 次日，她又召开职工大会，与职工们一起讨论出现质量不合格品的原因。 |
|---|---|
| English | The next day, she held another staff meeting to discuss with the staff the reasons for the unqualified quality products. |
| Labels | 举行;召集;举办;开;组织;主持;开展 |
| Dict | 举行;开;做(make) |
| Embedding | 开会研讨(Meeting discussion);局务(bureau affairs);党组会(party will);扩大会议(enlarged meeting); |
| | 审议(consider);召开大会(hold a convention);开会讨论(discussing in a Meeting);举行;现场会议(on site meetings);开完(after driving) |
| Bert | 召集;举行;开;组织;举办;参加(participate in);出席(attend);主持;进行(have);启动(firing) |
| ParaLS | 召集;举行;开;组织;举办;参加(participate in);开展;出席(attend);开起(start);主持 |
| Ensemble | 举行;召集;开;举办;出席(attend);组织;开展;参加(participate in);主持;开完(after driving) |
| **Inst. 2** | 次日，她又召开职工大会，与职工们一起讨论出现质量不合格品的原因。 |
| English | The next day, she held another staff meeting to discuss with the staff the reasons for the unqualified quality products. |
| Labels | 探讨;谈论;研讨;议论;商讨;商议;研究;协商;谈谈 |
| Dict | 探讨;谈论;商议;议论;研讨;谈谈;审议(consider);座谈(have an informal discussion);讲论(lectures);议事(proceedings) |
| Embedding | 开会研讨(meeting discussion);深入探讨(in-depth discussion);争辩(argue);议(opinion);争论不休(an endless debate); |
| | 开会讨论(discussing in a meeting);争论(debate);争论中(in dispute);商讨;辩论(debate) |
| Bert | 探讨;商讨;谈论;商议;研究;商量(discuss);议论;分析(analysis);研讨;争论(debate) |
| ParaLS | 探讨;商讨;谈论;商议;研究;商量(discuss);分析(analysis);议论;商谈(negotiate);协商 |
| Ensemble | 探讨;谈论;商议;议论;商讨;商量(discuss);研讨;研究;分析;协商 |
| **Inst. 3** | 在实践中，我们体会到，企业只有把产品质量看作自己的 生命，才能振兴。 |
| English | In practice, we have learned that a company can only be revitalized if it considers product quality to be its life. |
| Labels | 感受;体悟;体验;领悟;领会;认识;体味;感悟;意识 |
| Dict | 体验;感受;体味;经验(experience);认知(cognition);回味(aftertaste);咀嚼(chew);心得(experience);吟味(recite with relish) |
| Embedding | 理解(understand);亲身(in person);深切体会(deep experience);体会出(experience);深深感到(deeply felt);体验; |
| | 领会到(understand);领略(appreciate);领悟;感悟 |
| Bert | 体悟;认识;体验;感受;了解(understand);领悟;感悟;意识;领会;学习(study) |
| ParaLS | 认识;体悟;体验;意识;了解(了解);感受;领悟;感觉(Feelings);感悟;领会 |
| Ensemble | 体验;感受;体悟;领会;感悟;领会;认识;了解(understand);意识;体味 |
| **Inst. 4** | 由于人文地理的原因，白沟人经商确有优势。 |
| English | Due to cultural and geographical reasons, the people of Baigou do have advantages through negotiation. |
| | sewing machines, so the market boom concealed a potential quality crisis. |
| Labels | 缘故;因素;缘由;原故;主因;原由;原委 |
| Dict | 缘故;原故;缘由;原由;因;原委;由来(origin);由(cause);故(event);因由(cause) |
| Embedding | 种种原因(various reasons);究其(investigate it);诱因(incentive);根源(root);缘由;起因(cause);间接原因(indirect causes);因素; |
| | 直接原因(direct cause);决定因素(determinant) |
| Bert | 缘故;因素;影响(effect);优势(advantage);关系(relationship);理由(reason);特殊(special);原故;特点(characteristic);缘由 |
| ParaLS | 缘故;因素;影响(effect);优势(advantage);关系(relationship);理由(reason);特殊(special);特点(characteristic);考虑(consider);原故 |
| Ensemble | 缘故;原故;因素;理由(reason);影响(effect);优势(advantage);关系(relationship);缘由;特殊(special);特点(characteristic) |
| **Inst. 5** | 白沟市场一下就冷清下来，我的买卖也跟着衰了。 |
| English | The Baigou market immediately cooled down, and my business also declined. |
| Labels | 沉寂;萧条;冷淡;寂静;冷落;冷清清;清淡;冷冷清清 |
| Dict | 冷静(calm);冷落;清冷(chilly);萧条;冷冷清清;冷清清;落寞(lonely);空荡荡(empty);萧索(desolate);门可罗雀(there are very few people) |
| Embedding | 清静(quiet);静寂(silence);冷落;荒凉(desolate);凄凉(dreariness);落寞(lonely);冷淡;空空荡荡;冷清清;凄冷(cold) |
| Bert | 冷淡;寂静;冷(cold);冷静(calmness);清淡;平静(calmness);沉寂;寒冷(cold);清静(quiet);安静(quiet) |
| ParaLS | 冷淡;寂静;冷(cold);冷静(calmness);清淡;冷落;清冷(chilly);黯淡(dim);冷却(cooling);平静(calmness) |
| Ensemble | 冷淡;寂静;冷静(calmness);冷落;清冷(chilly);清淡;冷(cold);黯淡(dim);沉寂;平静(calmness) |

Table 5: The top 10 substitutes of five instances in the News subset of CHNLS using LS methods. The target word is marked in blue, and the substitutes in labels are marked in red.

| Inst. 1 | 他上身赤裸，下身套着一条皮裤，想是睡梦中得到讯息，赶来求情。 |
|---|---|
| English | He was naked and had a pair of leather pants on underneath, so he got the message in his sleep and came to plead for mercy. |
| Labels | 消息;信;信息;信号;情报;讯号;资讯 |
| Dict | 消息;资讯;快讯(news flash);情报;新闻(news);讯(interrogate);谍报(spy report) |
| Embedding | 小道消息(grapevine);消息;检察信息(Inspection Information);无用信息(Useless information);短消息(Short Message);死讯(News of death);商品信息(Product Information);病毒信息(Virus Information);传送(Transmission);传递信息(Delivering information) |
| Bert | 消息;信息;资讯;通知(Notification);信号;线索(Clues);情报;音讯(Audio);短信(SMS);新闻(News) |
| ParaLS | 信息;消息;资讯;通知(Notification);信号;线索(Clues);音讯(Audio);情报;短信(SMS);新闻(news) |
| Ensemble | 消息;资讯;信息;信号;情报;通知(Notification);讯号;音讯(Audio);新闻(news);呼唤(call out) |

| Inst. 2 | 郭靖急扑后帐，左手扯住帐幕用力拉扯，将半座金帐拉倒，罩在诸将头上。" |
|---|---|
| English | Guo Jing rushed towards the back tent, grabbed it with his left hand and pulled it hard, pulling down half of the golden tent and covering it over the heads of the generals. |
| Labels | 使劲;用劲;强力;加力;竭力;大力 |
| Dict | 大力;努力(Effort);使劲;极力(Make an effort);奋力(Endeavor);用劲;尽力(Try your best);一力(One Power);全力(Full strength);拼命(do one's utmost) |
| Embedding | 擂打(beat);用手掌(Use the palm of your hand);要用力(To force);撅住(Snap);使劲;撬着(pry);死劲(Dead weight);住(lay your hand upon);狠命(Tough life);搓擦(Rubbing) |
| Bert | 努力(Effort);大力;使劲;强力;用劲;尽力(Try your best);拼命(do one's utmost);轻轻(gently);猛烈(fierce);狠狠(Ruthlessly) |
| ParaLS | 大力;努力(Effort);使劲;强力;极力(Make an effort);奋力(Endeavor);用劲;加力(Powerful) |
| Ensemble | 使劲;用劲;大力;极力(Make an effort);尽力(Try your best);轻轻(gently);努力(Effort);强力;奋力(Endeavor);猛烈(Fierce) |

| Inst. 3 | 郭靖急扑后帐，左手扯住帐幕用力拉扯，将半座金帐拉倒，罩在诸将头上。" |
|---|---|
| English | Guo Jing rushed towards the back tent, grabbed it with his left hand and pulled it hard, pulling down half of the golden tent and covering it over the heads of the generals. |
| Labels | 拉拽;扯;撕扯;拽;拉;拉住;牵扯 |
| Dict | 拉;拉拉(Lara);拉开(Pull away);扯;牵扯;拽;拉长(Elongation);拉杆(Tie Rod);攀扯(Climbing);直拉(Straight pull) |
| Embedding | 扯下去(Pull it down);厮打(fight together);扯拉(Tug);推打(push and beat);扯住(Tug);揪扯(tug at);拉拉扯扯(pulling and tugging);扯碎(tear apart);扯断(Tear off);扭去(twist off) |
| Bert | 拉拽;拉;拉拉(Lara);撕扯;拉伸(Stretching);拉起(Pull up);拉住;拉开(Pull away);拉拢(rope in);拉紧(Tensioning) |
| ParaLS | 拉拽;拉;拉拉(Lara);扯拉(tug at);撕扯;拉伸(Stretching);扯拉(Tug);牵扯;拉起(Pull up);拉住 |
| Ensemble | 牵扯;扯;拉拽;拉;拉拉(Lara);撕扯;扯拉(Tug);拽;扯扯(tug at);拉住 |

| Inst. 4 | 又行十余里，两人下马互拜，搂抱了一会，洒泪而别。 |
|---|---|
| English | After walking for more than ten miles, the two dismounted and worshipped each other, hugging each other for a while, and parting in tears. |
| Labels | 拥抱;搂;抱抱;偎抱;揽 |
| Dict | 拥抱;搂;抱抱;抱(hug);揽 |
| Embedding | 搂搂(snuggle);狂吻(Kissing furiously);拥吻(Kissing);抚摸(Stroking);搂着(Cuddle up);抱腰(lap);亲热(affectionate);踮着(stand on tiptoe);搂住(hold in one's arms);紧抱(Hold tightly) |
| Bert | 拥抱;搂;抱;抱抱;搂住(hold in one's arms);相拥(embrace each other);亲吻(kiss);偎抱;吻(lips);拥(embrace) |
| ParaLS | 拥抱;搂;抱抱;搂住(hold in one's arms);相拥(embrace each other);亲吻(kiss);怀抱(embrace);偎抱;拥吻(Smooch) |
| Ensemble | 搂;拥抱;抱抱;抱(carry in breast);偎抱;亲吻(kiss);拥吻(Smooch);搂搂(snuggle);抚摸(Stroking);搂住(hold in one's arms) |

| Inst. 5 | 他故意替哲别掩饰，以免成吉思汗知晓内情。 |
|---|---|
| English | He deliberately covered up for Zhebei so that Genghis Khan would not know the inside story. |
| Labels | 有意;蓄意;成心;存心;特意;刻意 |
| Dict | 有意;蓄意;假意(hypocrisy);存心;有心(intend to);有意识(consciously);成心;特此(hereby);故(event);明知故犯(Knowingly committing a crime);偷偷(secretly);造谣生事(spread rumours to create trouble);成心;假意(pretend);反诬(make a false countercharge);故意杀人罪(Intentional Homicide);明知(Knowingly);诬赖(inculpate);故弄(Plague);卖弄玄虚(make a mystery of something) |
| Bert | 刻意;特意;设法(try to);有意;蓄意;假装(pretend);暗中(secretly);试图(try);打算(intend);暗暗(secretly) |
| ParaLS | 刻意;设法(try to);特意;有意;假装(pretend);暗中(secretly);蓄意;打算(intend);试图(try);暗暗(secretly) |
| Ensemble | 刻意;特意;有意;蓄意;假意(pretend);假装(pretend);存心;想(think);决定(decision);偷偷(secretly) |

Table 6: The top 10 substitutes of five instances in the Novel subset of CHNLS using LS methods. The target word is marked in blue, and the substitutes in labels are marked in red.