# OpenReview forum: "Chinese Lexical Substitution: Dataset and Method"
_EMNLP/2023/Conference — EMNLP 2023 Main_

### Official Review · Reviewer_24p3 · 2023-08-04

**Soundness:** 3

**Excitement:**

3: Ambivalent: It has merits (e.g., it reports state-of-the-art results, the idea is nice), but there are key weaknesses (e.g., it describes incremental work), and it can significantly benefit from another round of revision. However, I won't object to accepting it if my co-reviewers champion it.

**Paper Topic And Main Contributions:**

The paper proposes a novel  large-scale Chinese lexical substitution (LS) dataset created by human-machine collaboration. The key contributions are:

- Construction of a large-scale Chinese LS dataset, CHNLS, containing 33,695 instances across 3 text genres. Significantly larger than prior English datasets.
- Presentation of 4 LS methods, including dictionary, embedding, BERT and paraphraser-based approaches.
- An ensemble method combining the 4 approaches that outperforms individual methods on CHNLS evaluation.
- Quantitative and qualitative analysis showing the high coverage and quality of CHNLS.

**Questions For The Authors:**

- Did you apply any quality assurance measures been taken during the data annotation process? How is the consistency of the annotators' work ensured?
- It's recommended to increase the diversity of data to make this dataset more effective and compelling

**Reasons To Accept:**

- Addresses the lack of Chinese LS datasets and enables future research for this under-studied language.
- The collaborative annotation approach is creative, efficient and results in higher coverage compared to solely human annotation. Could inform future dataset creation.
- Comprehensive experiments demonstrate the utility of the dataset and the effectiveness of the proposed ensemble method. Thorough quantitative and qualitative analysis.

**Reasons To Reject:**

- While larger than prior datasets, CHNLS still only covers 3 genres of Chinese text. More diversity could be beneficial.
- Some subjectivity and noise inevitable during human evaluation of machine-generated substitutes. Inter-annotator agreement is unclear.
- Ensemble approach is relatively simple. More sophisticated methods could be explored for combining multiple LS techniques.
- Limited analysis of how well methods generalize across the 3 genres. More cross-genre evaluation would be informative.
- The dataset quality should be improved. For example, in wiki_test_gold.txt, line 1, the second substitute "既" is in correct. The authors should do double check to make sure the substitute is correct.

**Reproducibility:**

5: Could easily reproduce the results.

**Reviewer Confidence:**

4: Quite sure. I tried to check the important points carefully. It's unlikely, though conceivable, that I missed something that should affect my ratings.

---

> ### Author Rebuttal · Authors · 2023-08-27
>
> ***Question: Did you apply any quality assurance measures been taken during the data annotation process? How is the consistency of the annotators' work ensured?***
>
> **Response**:  Thank you for your valuable comment. We used common agreement metrics such as Cohen's Kappa [1] and Fleiss' Kappa [2] to quantify the level of agreement among annotators. Cohen's Kappa measures agreement between two raters, and Fleiss’ Kappa can be used for measuring agreement between multiple raters. The Kappa result be interpreted as follows: values ≤ 0 as indicating no agreement and 0.01–0.20 as none to slight, 0.21–0.40 as fair, 0.41– 0.60 as moderate, 0.61–0.80 as substantial, and 0.81–1.00 as almost perfect agreement.
>
> Table 1 lists the agreement scores for three annotators. Specifically, we calculated Fleiss' Kappa for our dataset, yielding a value of 0.594. This statistic underscores a substantial level of agreement among our human annotators, reaffirming the consistency and reliability of the annotations. We will ensure to include these statistics and details in the revised version of the paper to provide a comprehensive understanding of our evaluation process.
>
>
> Table 1. Cohen’s kappa agreement scores for pairs of annotators and Fleiss’ kappa agreement for three annotators.
> | Cohen’s (A1–A2) | Cohen’s (A1–A3) | Cohen’s (A2–A3) | Fleiss’ kappa |
> |-----------------|-----------------|-----------------|---------------|
> | 0.598           | 0.614           | 0.572           | 0.594        |
>
>
>
> ***Question:  It's recommended to increase the diversity of data to make this dataset more effective and compelling***
>
> ***Response***: Thank you for your thoughtful comment and recommendation. We appreciate your feedback and understand the importance of data diversity in enhancing the effectiveness and appeal of our dataset. We are currently constructing LS datasets for two newly genres (legal and medical), and we will add them to the revised version of the paper.
>
>
>
> ***Question: The dataset quality should be improved. For example, in wiki_test_gold.txt, line 1, the second substitute "既" is in correct. The authors should do double check to make sure the substitute is correct.***
>
> ***Response***: We apologize for the confusion. Indeed, the substitute "既" is considered to be correct in our dataset. The '2' after "既" in wiki_test_gold.txt means that two annotators of three annotators believe that this substitute word is correct. We will provide a detailed introduction to the dataset in the final paper.
>
> [1]  J. Cohen. A coefficient of agreement for nominal scales. Educational and Psychological Measurement, 1960, 20:37–46.
> [2]  J.L. Fleiss. Measuring nominal scale agreement among many raters. Psychological Bulletin, 1971, 76:378–382.

---

### Official Review · Reviewer_o7zy · 2023-08-11

**Soundness:** 4

**Excitement:**

4: Strong: This paper deepens the understanding of some phenomenon or lowers the barriers to an existing research direction.

**Paper Topic And Main Contributions:**

The paper “Chinese Lexical Substitution: Dataset and Method” presents a novel approach for the creation of annotated Lexical Substitution datasets, focusing on the Chinese language to address the lack of an existing Chinese data source for LS. In addition, the authors propose a new ensemble method, combining four classes of LS methods. In doing so, the authors evaluate both the dataset they have created, as well as the effectiveness of their ensemble approach.

**Questions For The Authors:**

Question A: can you explain why some translations are left out of the tables?

Question B: were any inter-annotator agreement statistics calculated?

**Reasons To Accept:**

The contributions presented by the authors in this paper are many.

Firstly, the authors highlights shortcomings of existing LS datasets, particularly pointing to their relatively small scale and lack of coverage. While this has been pointed out in the literature before (e.g., SwordS), the authors justify that the issues still persist. Moreover, the lack of LS resources for the Chinese language is highlighted.

The new annotation scheme presented is highly original. Motivated by the issue of small-scale datasets, the authors propose a method that allows for larger-scale corpus creation via the automation of existing methods. The quality of these outputs is ensured by the placement of human-made decisions at the end of the pipeline. The creation and execution of this process is well-described and transparent.

The introduction of the CHNLS dataset is likewise a great contribution. This is supplemented by descriptive statistics of the dataset, as well as a discussion of the evaluation that was run to measure the quality and coverage.

The experiments run using the novel ensemble method are demonstrated by using well-known LS metrics from LS07, as well as the presentation of illustrative examples.

The paper is concluded by a meaningful discussion of the results, particularly in comparison to existing datasets and methods.


**Reasons To Reject:**

Besides the occasional grammatical or stylistic errors (some highlighted below), there are only a few points of weakness exhibited by the paper.

For example, the details of the ensemble method are a bit lacking. The exact construction of the ensemble, as well as the scheme used to weigh the individual methods’ scores, is not fully elucidated.

In addition, the proposed ensemble is only evaluated on the newly created dataset, making it hard to compare against other existing methods, such as those that have been evaluated on English data.

Finally, while the top 10 substitutes tables are interesting, they are quite cluttered and slightly unclear. Who determined the substitutes in red? Moreover, it is difficult to interpret the results where no translation is provided.

**Reproducibility:**

5: Could easily reproduce the results.

**Reviewer Confidence:**

3: Pretty sure, but there's a chance I missed something. Although I have a good feel for this area in general, I did not carefully check the paper's details, e.g., the math, experimental design, or novelty.

**Typos Grammar Style And Presentation Improvements:**

Line 38: this and the following sentence should be one sentence, yet they are split.

455: belonging -> belong

The naming of Appendix B presumably needs to be changed from the default.

---

> ### Author Rebuttal · Authors · 2023-08-27
>
> **Question A: can you explain why some translations are left out of the tables?**
>
> **Response**: Thank you. The words in labels and the words in red are all correct substitutions. Their translations are the same as the translations of the target words (shown in blue font), so the translations of these words are not provided. In the revised version, we will provide explanations in the captions to offer further clarity.
>
> **Question B: were any inter-annotator agreement statistics calculated?**
>
> **Response**:  Thank you for your valuable comment. We used common agreement metrics such as Cohen's Kappa [1] and Fleiss' Kappa [2] to quantify the level of agreement among annotators. Cohen's Kappa measures agreement between two raters, and Fleiss’ Kappa can be used for measuring agreement between multiple raters. The Kappa result be interpreted as follows: values ≤ 0 as indicating no agreement and 0.01–0.20 as none to slight, 0.21–0.40 as fair, 0.41– 0.60 as moderate, 0.61–0.80 as substantial, and 0.81–1.00 as almost perfect agreement.
>
> Table 1 lists the agreement scores for three annotators. Specifically, we calculated Fleiss' Kappa for our dataset, yielding a value of 0.594. This statistic underscores a substantial level of agreement among our human annotators, reaffirming the consistency and reliability of the annotations. We will ensure to include these statistics and details in the revised version of the paper to provide a comprehensive understanding of our evaluation process.
>
>
> Table 1. Cohen’s kappa agreement scores for pairs of annotators and Fleiss’ kappa agreement for three annotators
> | Cohen’s (A1–A2) | Cohen’s (A1–A3) | Cohen’s (A2–A3) | Fleiss’ kappa |
> |-----------------|-----------------|-----------------|---------------|
> | 0.598           | 0.614           | 0.572           | 0.594         |
>
> [1]  J. Cohen. A coefficient of agreement for nominal scales. Educational and Psychological Measurement, 1960, 20:37–46.
>
> [2]  J.L. Fleiss. Measuring nominal scale agreement among many raters. Psychological Bulletin, 1971, 76:378–382.

---

### Official Review · Reviewer_tGuY · 2023-08-11

**Paper Topic And Main Contributions:** 1. This paper introduces CHNLS, a ben…
**Soundness:** 2

**Excitement:**

3: Ambivalent: It has merits (e.g., it reports state-of-the-art results, the idea is nice), but there are key weaknesses (e.g., it describes incremental work), and it can significantly benefit from another round of revision. However, I won't object to accepting it if my co-reviewers champion it.

**Missing References:**

No

**Questions For The Authors:**

Question A: Have you tried using chatGPT or other LLMs to produce data?



**Reasons To Accept:**

1.The first benchmark for Chinese lexical substitution (LS).
2. The mainstream models have undergone comprehensive evaluation.
3. being clear and well-written



**Reasons To Reject:**

1. Lack elucidation of certain pertinent indicators. such as "best-m," "oot," and "oot-m,"
2. Chinese translator to translate english sentences into Chinese may introduce noise.
3. Lack  baseline test for  LLMs.

**Reproducibility:**

3: Could reproduce the results with some difficulty. The settings of parameters are underspecified or subjectively determined; the training/evaluation data are not widely available.

**Reviewer Confidence:**

3: Pretty sure, but there's a chance I missed something. Although I have a good feel for this area in general, I did not carefully check the paper's details, e.g., the math, experimental design, or novelty.

---

> ### Author Rebuttal · Authors · 2023-08-27
>
> ***Question A: Have you tried using chatGPT or other LLMs to produce data?***
>
> ***Response***: Thank you for your valuable comment. Indeed, we did explore the possibility of utilizing ChatGPT for substitution generation in our study. However, two primary reasons prompted us not to employ ChatGPT or other LLMs:
>
> (1) Following an analysis of numerous samples, we observed that the performance of ChatGPT for Chinese lexical substitution task falls short compared to that of BERT and ParaLS. The generated substitutes from ChatGPT consistently overlap with those generated by the four methods we have presented.
>
> (2) It is a high cost associated with constructing a large-scale dataset with 36,000 instances using ChatGPT. Considering we already choose four lexical substitution methods based on different semantic resources, ChatGPT as the five method is not very necessary.
>
> ***Question: Lack baseline test for LLMs.***
>
> ***Response***: Drawing from your comments, we conducted tests employing two LLMs as the baselines for comparison. In addition to selecting ChatGPT utilizing the “gpt-35-turbo” API, we also opted for an open-sourced LLM ChatGLM based on “GLM-130B” API， which is very good at handling Chinese. The two LLMs in the experiments are based one-shot learning. We translated the prompt used by Aumiller et al. (2023)[1] as our prompt. More details about the prompt can be found in Table 1.
>
> Table 2 displays the performance of all methods on the various metrics, where the results of LLMs (ChatGPT and ChatGLM) are shown in bold. We found that two single methods (BERT and ParaLS) outperform ChatGPT and ChatGLM. Ensemble surpasses the individual LS methods on all metrics with statistical significance. We will include the two LLMs in the revised version.
>
>
> Table 1. One-shot Prompt for Chinese lexical substitution
> |                           Prompt                           |
> |:----------------------------------------------------------:|
> | 上下文: 如每户农民都在地块上打井，势必造成资金的浪费。     |
> | 问题: 列出十个可以替换上面句子中"势必"的词(以逗号分隔)     |
> | 回答: 必然,必将,必定,肯定,一定,势将,定会,必,将,会          |
> | 上下文: {原始句子}                                         |
> | 问题: 列出十个可以替换上面句子中"{目标词}"的词(以逗号分隔) |
> | 回答:                                                      |
>
> Table 2. Evaluation results by adding two methods using LLMs (ChatGLM and ChatGPT)
> | Dataset |   Method  |  best | best-m |  oot  | Oot-m |  P@1  |
> |:-------:|:---------:|:-----:|:------:|:-----:|:-----:|:-----:|
> |   Wiki  | Dict      |  16.4 |  24.0  |  39.0 |  41.4 |  51.2 |
> |         | Embedding |  13.6 |  21.2  |  46.4 |  55.1 |  41.1 |
> |         | BERT      |  16.9 |  25.7  |  53.9 |  58.4 |  50.0 |
> |         | ParaLS    |  17.0 |  25.7  |  54.1 |  59.8 |  48.5 |
> |         | Ensemble  |  18.7 |  29.4  |  66.6 |  73.4 |  57.0 |
> |         | **ChatGLM**   |  **12.2** |  **17.9**  |  **34.1** |  **39.3** |  **35.4** |
> |         | **ChatGPT**   |  **15.8** | **26.1**  |  **39.4** |  **47.5** |  **44.5** |
> |   News  | Dict      |  13.2 |  24.5  |  37.3 |  41.6 |  65.9 |
> |         | Embedding |  12.9 |  23.6  |  45.5 |  56.5 |  55.8 |
> |         | BERT      |  17.2 |  32.9  |  62.1 |  66.6 |  80.9 |
> |         | ParaLS    |  17.3 |  34.5  |  66.3 |  71.2 |  79.5 |
> |         | Ensemble  | 18.4 |  36.4  |  66.3 |  71.2 |  79.5 |
> |         | **ChatGLM**   | **9.9**  |  **15.5**  | **33.8** |  **42.1** |  **48.1** |
> |         | **ChatGPT**   |  **14.0** |  **28.3**  |  **36.6** |  **43.5** |  **62.0** |
> |  Novel  | Dict      |  12.5 |  17.0  |  41.3 |  40.0 |  54.5 |
> |         | Embedding |  14.0 |  16.8  |  46.5 |  46.4 |  46.1 |
> |         | BERT      |  13.8 |  18.6  |  44.9 |  47.8 |  54.2 |
> |         | ParaLS    |  13.8 |  18.6  |  44.9 |  47.8 |  54.2 |
> |         | Ensemble  | 18.1 |  26.7 | 65.8 | 70.3 | 64.3|
> |         | **ChatGLM**   |  **10.3** |  **11.0**  |  **32.3** |  **34.0** |  **38.1** |
> |         | **ChatGPT**   |  **12.4** |  **14.7**  |  **35.4** |  **37.5** |  **45.5** |
>
> [1] D. Aumiller, M. Gertz. UniHD at TSAR-2022 Shared Task: Is Compute All We Need for Lexical Simplification. arXiv preprint arXiv:2301.01764, 2023.

---

### Meta-Review · Area_Chair_QRzK · 2023-09-18

**Recommendation:** 5

**Metareview:**

The paper presents a new large-scale Chinese lexical substitution (LS) dataset for 3 genres News, Novels, and Wikipedia, from different sources. This resource fills a gap in Chinese NLP. This dataset is meant to serve as the first benchmark for Chinese lexical substitution (LS), and thus has a great potential for the community. Together with the dataset, the intriguing contribution of this paper is a novel ensemble method for producing it, which combines four distinct LS techniques with the goal to enhance the diversity of substitutes. Evaluation has been conducted that positively demonstrates both the quality of the dataset and the effectiveness of their ensemble approach. The insights from this study could be of great value esp. for future similar works. Although the paper has many strengths (see pros. for other positive aspects not mentioned above), it also shows a few flaws which would increase the impact of presentation/publication (see cons) if amended. The following summarizes the most salient. Please refer to original reviews for improvement suggestions.

**Other Pros**

- Dataset and method are clearly described; descriptive statistics of the dataset are included;

- The paper is very well framed and organized, the experiments clearly motivated and argumentation effective and exhaustive notwithstanding the page number limitation;

- Related literature is well covered;

- shortcomings of existing LS dataset are highlighted;

- It introduces an intriguing novel hybrid approach for the creation of annotated Lexical Substitution datasets, which allows for larger-scale corpus creation confronted with human-only labeled data;

- Includes a meaningful comparison to existing datasets and methods;

- Development of the first paraphrase corpus for Chinese for training a paraphrase-based method to be used in the ensemble.



**Cons**

- lack of discussion (or acknowledgement as a limitation) of the potential noise introduced by automatic translation;

- missing IAA calculation; authors should integrate the data reported in their response;

- lack of cross-genre and cross-language evaluation or a discussion on the generalizability of the methods;

---

### Decision · Program_Chairs · 2023-10-07

**Decision:**

Accept-Main

**Comment:**

The paper presents a new large-scale Chinese lexical substitution (LS) dataset for 3 genres News, Novels, and Wikipedia, from different sources. This resource fills a gap in Chinese NLP. This dataset is meant to serve as the first benchmark for Chinese lexical substitution (LS), and thus has a great potential for the community. Together with the dataset, the intriguing contribution of this paper is a novel ensemble method for producing it, which combines four distinct LS techniques with the goal to enhance the diversity of substitutes. Evaluation has been conducted that positively demonstrates both the quality of the dataset and the effectiveness of their ensemble approach. The insights from this study could be of great value esp. for future similar works. Although the paper has many strengths (see pros. for other positive aspects not mentioned above), it also shows a few flaws which would increase the impact of presentation/publication (see cons) if amended. The following summarizes the most salient. Please refer to original reviews for improvement suggestions.

**Other Pros**

- Dataset and method are clearly described; descriptive statistics of the dataset are included;

- The paper is very well framed and organized, the experiments clearly motivated and argumentation effective and exhaustive notwithstanding the page number limitation;

- Related literature is well covered;

- shortcomings of existing LS dataset are highlighted;

- It introduces an intriguing novel hybrid approach for the creation of annotated Lexical Substitution datasets, which allows for larger-scale corpus creation confronted with human-only labeled data;

- Includes a meaningful comparison to existing datasets and methods;

- Development of the first paraphrase corpus for Chinese for training a paraphrase-based method to be used in the ensemble.



**Cons**

- lack of discussion (or acknowledgement as a limitation) of the potential noise introduced by automatic translation;

- missing IAA calculation; authors should integrate the data reported in their response;

- lack of cross-genre and cross-language evaluation or a discussion on the generalizability of the methods;